# Characterization of Thermomechanical Boundary Conditions of a Martensitic Steel for a FAST Forming Process

**Xiaochuan Liu** [1] , **Xiao Yang** [1], **Yuhao Sun** [1], **Denis J. Politis** [2], **Ken-ichiro Mori** [3] **and Liliang Wang** [1,*]

1    Department of Mechanical Engineering, Imperial College London, London SW7 2AZ, UK;
     xiaochuan.liu14@imperial.ac.uk (X.L.); x.yang17@imperial.ac.uk (X.Y.); yuhao.sun14@imperial.ac.uk (Y.S.)
2    Department of Mechanical and Manufacturing Engineering, University of Cyprus, Nicosia 20537, Cyprus;
     politis.denis@ucy.ac.cy
3    Department of Mechanical Engineering, Toyohashi University of Technology, Toyohashi Aichi 441-8580,
     Japan; mori@plast.me.tut.ac.jp
*    Correspondence: liliang.wang@imperial.ac.uk; Tel.: +44-20-7594-3648

**Abstract:** The present work characterized and modelled the interfacial heat transfer coefficient and friction coefficient of a non-alloy martensitic steel, for a novel Fast light Alloy Stamping Technology (FAST) process. These models were validated through temperature evolution, thickness distribution and springback measurements on experimentally formed demonstrator components, which were conducted on a pilot production line and showed close agreement, with less than 10% variation from experimental results. The developed models and finite element simulations presented in this work demonstrate that non-isothermal processes can be precisely simulated with implementation of the accurate thermomechanical boundary conditions.

**Keywords:** interfacial heat transfer coefficient; friction coefficient; Fast light Alloy Stamping Technology; non-alloy steel; FEM

## 1. Introduction

In recent years, hot and warm stamping technologies have been widely applied to the forming of sheet metals, in which a sheet blank is first heated to elevated temperatures, transferred to a press, and then deformed and simultaneously quenched within tooling at, or slightly above, room temperature. Mori et al. [1] applied hot and warm stamping technologies to form high-strength steels, while El Fakir et al. [2] used a hot stamping process (HFQ®) to form complex-shaped aluminum alloy components. It has also been proven that titanium alloys could be formed under hot/warm stamping conditions to produce complex geometries [3]. These promising technologies are significantly beneficial for improving the formability of materials, while reducing the springback of formed components, which has been of significant interest to academia and industry [4]. The novel Fast light Alloy Stamping Technology (FAST) is one such forming process offering fast heating and quenching, which enables the forming of complex-shaped components with dramatically reduced cycle times [5]. In order to reduce experimental efforts, finite element (FE) simulation is generally performed, to optimize the processing window and evaluate the post-form quality of the formed component [6]. However, non-isothermal forming processes are particularly challenging, as the FE simulation demands accurate assignments of thermomechanical boundary conditions, including the friction coefficient and interfacial heat transfer coefficient (IHTC) [7].

The friction coefficient under hot/warm stamping conditions is critical to describing the contact conditions at the tool–workpiece interface [8], as this affects the drawability of the material and the springback of the formed component [9]. It has been found that the friction coefficient decreased with increasing temperature, which is typically evaluated by either pin-on-disc/strip or drawing tests [10–12]. Lubricants are widely used in metal forming processes to reduce the friction forces at the tool/workpiece interfaces. However, lubricant breakdown may occur during the forming process, causing abrupt increases of the friction coefficient [13]. Meanwhile, the IHTC determining the temperature field is of great importance in non-isothermal forming processes [14]. Similar to the friction coefficient, the IHTC is also affected by contact pressure and lubricant [15]. The combination of these parameters and their effects on the IHTC may influence the post-form strength of materials such as high-strength steels [16] and heat-treatable aluminum alloys [17], due to precipitates generated during quenching. Therefore, the modelling of the IHTC and friction coefficient are critical in determining the temperature field, thickness/thinning distribution, springback and post-form strength of formed components [18].

Non-alloy martensitic steel has been widely used for vehicle components due to its excellent strength-to-price ratio [19]. Due to the high strength and low ductility of this material, hot roll forming, bending and crash forming are applied to manufacture non-alloy martensitic steel components [20–22]. The FAST forming technology overcomes the limitation of low ductility while maintaining high strength, thereby enabling the manufacture of complex-shaped martensitic steel components. Furthermore, the cycle time has been shown to be significantly reduced compared to a press hardening process [23]. In order to accurately simulate the interaction between the thermal and mechanical fields of the FAST forming process, the thermomechanical boundary conditions of the non-alloy martensitic steel must be determined. Therefore, the present work has characterized and modelled the IHTC and friction coefficient for the non-alloy martensitic steel under the FAST forming conditions. The determined thermomechanical boundary conditions were subsequently implemented into the FE simulation of the FAST forming of a U-shaped demonstrator component, to simulate the temperature field, thickness distribution and springback, enabling the comparison with experimental results obtained from the FAST forming tests. As a result, an optimized FE simulation of the FAST forming processes was realized.

## 2. Experimental Procedures

The non-alloy martensitic steel used in the present research was provided by ThyssenKrupp (ThyssenKrupp AG, Essen, Germany), as 1.6 mm thick blank sheets in a grade of hot-rolled MS-W 900Y1180T with an electrogalvanized zinc coating, with its chemical composition shown in Table 1. In order to determine the IHTC and friction coefficient for the martensitic steel, the heat transfer test and friction test were conducted respectively under the FAST forming conditions. Subsequently, the martensitic steel was formed into a U-shaped component under the same conditions, of which the temperature field, thickness distribution and springback were measured by the dedicated equipment to verify the determined thermomechanical boundary conditions. A graphite-based lubricant supplied by Sovereign Lubricants Ltd. was studied in the present research with the test matrix shown in Table 2.

**Table 1.** The chemical composition of the supplied non-alloy martensitic steel and P20 tool steel.

| Non-Alloy Martensitic Steel | | | | | | | | | |
|---|---|---|---|---|---|---|---|---|---|
| **Element** | **C** | **Si** | **Mn** | **P** | **S** | **Al** | **Ti+Nb** | **Cr+Mo** | **B** | **Fe** |
| Wt % | 0.25 | 0.8 | 2.5 | 0.06 | 0.015 | 0.015–2.0 | 0.25 | 1.2 | 0.005 | Bal. |

| AISI P20 Tool Steel | | | | | | | |
|---|---|---|---|---|---|---|---|
| **Element** | **C** | **Cr** | **Mn** | **Mo** | **Ni** | **Si** | **S** | **Fe** |
| Wt % | 0.37 | 2.0 | 1.4 | 0.2 | 1 | 0.3 | 0.01 | Bal. |

**Table 2.** The test matrix to characterize the thermomechanical boundary conditions.

| Heat Transfer | Pressure | Temperature | Lubricant | Purpose |
|---|---|---|---|---|
| | 0.5–50 MPa | 400 °C | None | Effect of pressure on the interfacial heat transfer coefficient |
| | 0.5–50 MPa | 400 °C | Graphite | Effect of lubricant on the interfacial heat transfer coefficient |
| **Friction** | **Load** | **Temperature** | **Lubricant** | **Purpose** |
| | 5 N | 350–450 °C | None | Effect of temp. on the friction |
| | 5 N | 350–450 °C | Graphite | Effect of lubricant on the friction |

## 2.1. Heat Transfer Test

An inverse finite element simulation technique and a dedicated testing facility (IHTC-Mate) [14] were used to determine the IHTC for non-alloy martensitic steel. In each heat transfer test, a 120 × 10-mm rectangular-shaped martensitic steel sample was screwed on the test facility, which was assembled on a Gleeble machine, as shown in Figure 1. The martensitic steel was heated to 400 °C at a high rate of 50 °C/s, while the tooling made from P20 tool steel remained at ambient temperature. The chemical composition of the AISI type P20 tool steel was shown in Table 1 and the material properties shown in Table 3. When the target temperature was achieved, the hot martensitic steel sample was compressed by the tooling, with contacting surfaces of 50 × 25 mm at contact pressures ranging from 0.5 to 50 MPa, for 20 s, to identify the effect of contact pressure on the IHTC. K-type thermocouples were embedded on the mid-thickness of the martensitic steel sample through a drilling hole shown in Figure 1c, connecting to a data logger in the Gleeble machine, in order to accurately measure the temperature evolution of the sample at its middle point. A similar method was used in Section 2.3 to embed the thermocouples for measuring the temperature evolution of martensitic steel in a FAST forming process. To study the effect of lubricant on the IHTC, the graphite-based lubricant was applied on the contacting surfaces of the punch and die before compression.

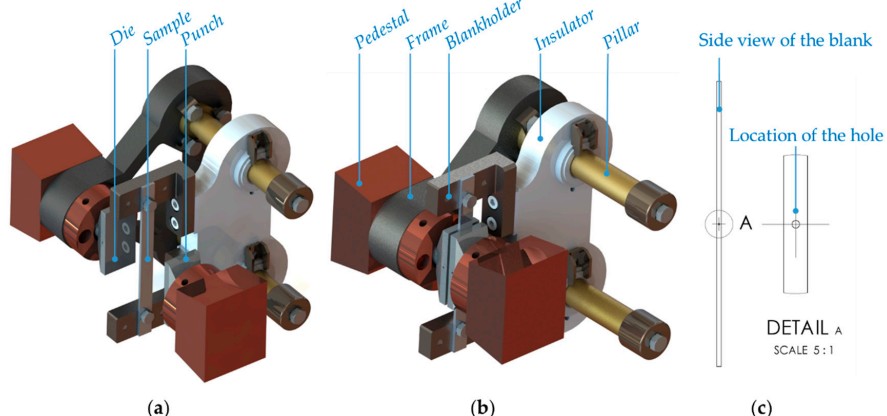

**Figure 1.** The heat transfer testing facility (IHTC-Mate) under (**a**) unloading conditions; (**b**) loaded conditions [14]; and (**c**) the drilling hole on the blank for embedding the thermocouples.

**Table 3.** The material properties of the non-alloy martensitic steel and AISI P20 tool steel.

| Material | Elastic Modulus | Density | Thermal Conductivity | Specific Heat | Expansion Coefficient |
|---|---|---|---|---|---|
| MS | 210 GPa | 7780 kg/m$^3$ | 24.5 kW/(m·K) | 431 J/(kg·K) | 11.1 μm/(m·K) |
| P20 | 205 GPa | 7850 kg/m$^3$ | 31.5 kW/(m·K) | 473 J/(kg·K) | 12.8 μm/(m·K) |

## 2.2. Friction Test

The friction tests were conducted on a pin-on-strip testing facility, as shown in Figure 2 [24]. A pin of spherical contact surface with a radius of 4 mm was made from P20 tool steel, while

a rectangular-shaped strip was made from the non-alloy martensitic steel. In each test, the martensitic steel strip was first heated to the target temperature. Subsequently, the cold P20 pin was activated to linearly slide on the strip at a load of 5 N for a sliding distance of 70 mm. Target temperatures of 350 °C, 400 °C and 450 °C were employed on the martensitic steel strip to characterize the effect of temperature on the friction coefficient. In addition to the dry sliding conditions, the graphite lubricant was applied on the P20 pin before sliding on the martensitic steel at elevated temperatures to characterize the effect of lubricant on the friction coefficient.

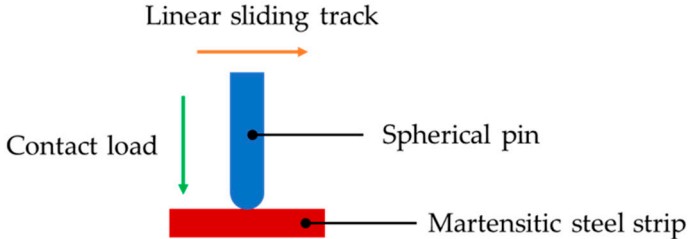

**Figure 2.** The schematic diagram of the friction test.

### 2.3. Forming Test

The FAST forming of U-shaped demonstrator components was conducted on a pilot production line shown in Figure 3 [25]. In a forming test, a 240 × 86 mm rectangular-shaped non-alloy martensitic steel blank was first transferred to a conduction heater by a conveyor, followed by fast-heating to the target temperature of 400 °C. Subsequently, the sample was rapidly transferred to a cold forming tool, which was composed of blankholders and a pair of punch and die made from P20 steel. After positioning, the cold punch was activated to move downwards at a forming speed of 250 mm/s to clamp the martensitic steel blank, with a constant blankholding force of 10 kN applied. The punch continually moved to the die to deform the martensitic steel and simultaneously quench the material to room temperature. After quenching, the tooling returned to its initial position, and the formed component was removed from the die. Sufficient lubricant was used on the tooling to avoid lubricant breakdown. Thermocouples were embedded into the martensitic steel to monitor temperature evolution during the full forming cycle. The thickness distribution of the central cross-section from the formed component was subsequently measured. Additionally, springback angles were measured by a LK G90C coordinate measurement machine to characterize the curls of the sidewall and flange respectively. These experimental results were then compared with the simulated results shown in Section 5.2.

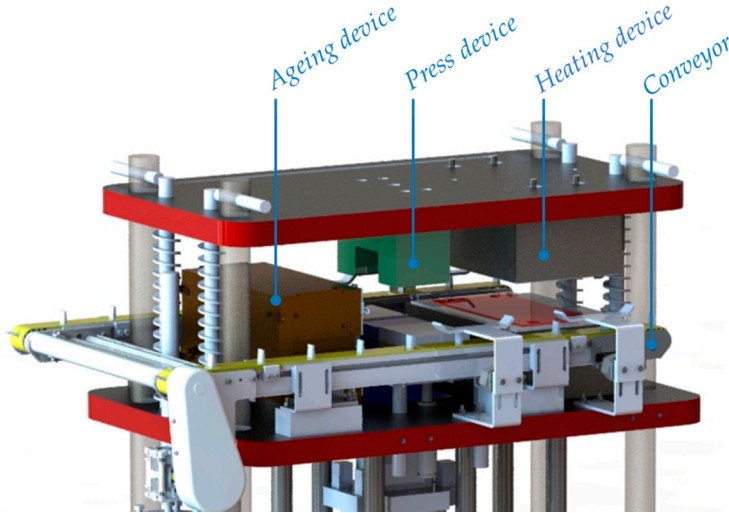

**Figure 3.** The schematic diagram of the pilot production line.

## 3. FE Simulation

### 3.1. Characterization of the IHTC by Using an Inverse Technique

The IHTC was characterized by using an inverse technique, through the comparison of the difference between the experimental and FE-simulated temperature histories. The FE simulation software PAM-STAMP (ESI Group, Paris, France) was used, which is tailored for sheet metal forming processes and can be used to access the interaction between the thermal and mechanical fields [26]. The FE model established in PAM-STAMP to predict the temperature history of the martensitic steel is shown in Figure 4a. The geometries and materials of the modelled components were identical to those assembled in the heat transfer experiment. Table 3 shows the material properties of the martensitic steel and P20 tool steel. It has been proven that the temperature distribution was convergent by using a mesh size of 2 mm [27]. Therefore, in the FE simulation of the heat transfer without deformation, quadrangle thermal shell elements with a size of 2 mm meshed the martensitic steel sample, punch/die and blankholder, resulting in their element numbers of 315, 325 and 634 respectively, as shown in Figure 4b,c. In addition, the 'hot stamping double-action validation' strategy was used to simulate the compression and quenching processes. As opposed to the heat transfer test, where a stable pressure was employed, a constant IHTC was assigned in the FE simulation to predict the heat transfer between the martensitic steel and tooling. Subsequently, the temperature histories of the martensitic steel were acquired by implementing various IHTC values, which were subsequently compared to the experimental results under different conditions. The best comparison between the experimental and FE-simulated results indicated that the stable IHTC assigned in the simulation was accurate. By using this inverse technique, the IHTC for the martensitic steel, at a range of pressures under the dry and lubricated-contact conditions, was characterized and then assigned in the subsequent simulation of the FAST forming of the U-shaped component to predict the accurate temperature distribution.

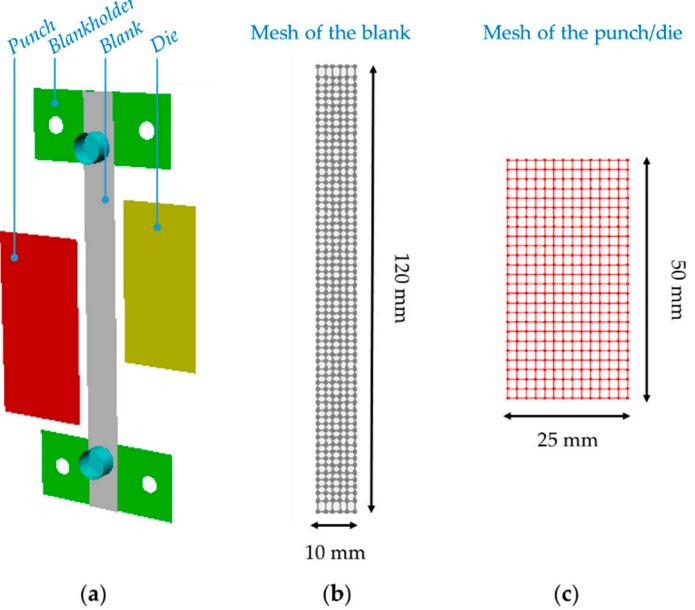

**Figure 4.** (**a**) The finite element models of the heat transfer to characterize the IHTC [14]; the mesh of (**b**) the martensitic steel blank and (**c**) the punch and die.

### 3.2. FE Simulation of a FAST Forming Process

An FE model was constructed to simulate the FAST forming of the U-shaped component, to obtain temperature evolution, thickness distribution and springback, which were used to verify the characterized thermomechanical boundary conditions of the martensitic steel. This FE model was

composed of a punch, die, blankholders and martensitic steel blank, to represent the press device in the pilot production line, as shown in Figure 5a.

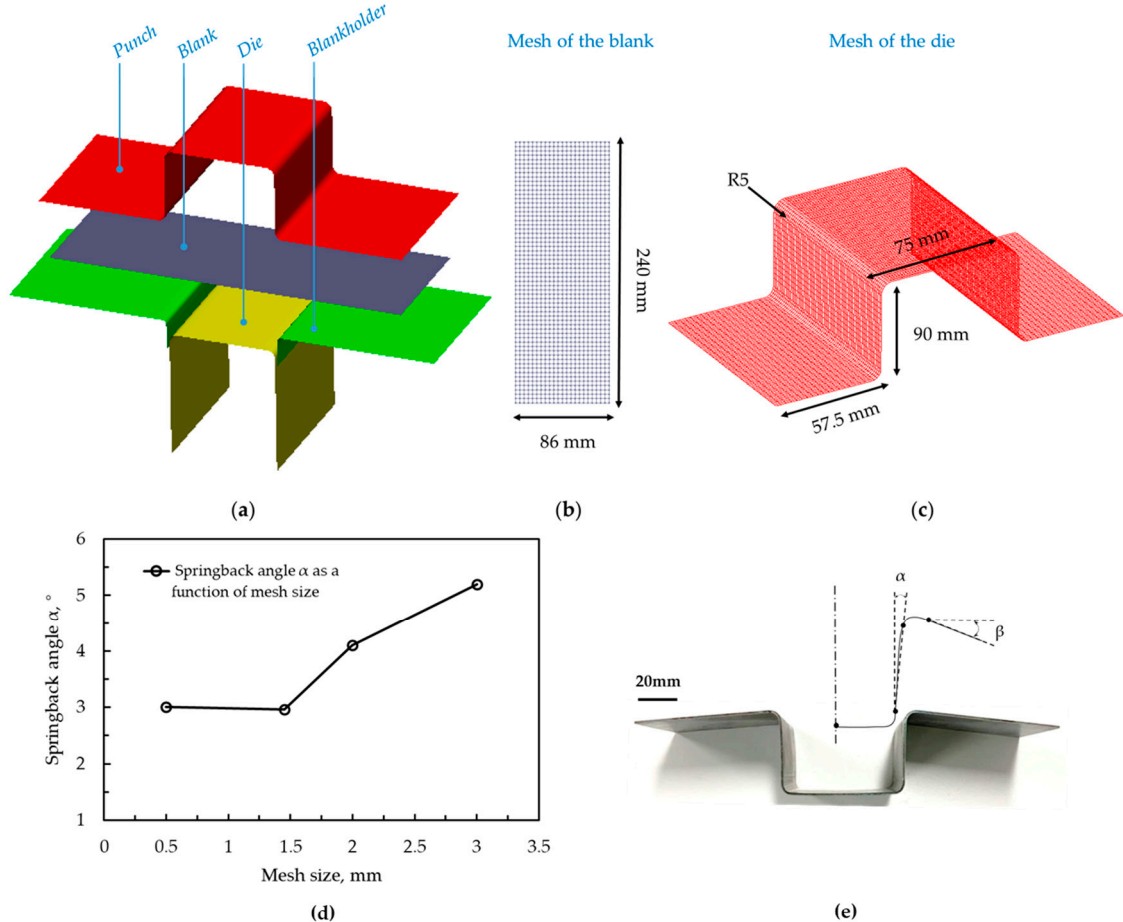

**Figure 5.** (**a**) The FE models of the FAST forming in the simulation software PAM-STAMP; the mesh of (**b**) the martensitic steel blank and (**c**) the die; (**d**) the mesh convergence study; (**e**) the springback angles of the formed component.

It was suggested in the literature that the mesh size should be 25% of the sum of the minimum fillet curvature radius (5 mm) and half-thickness of the blank (1.6 mm), for the simulation of springback [26]. Furthermore, a mesh size study, shown in Figure 5d, proved that the simulated springback angles reached convergence when the mesh size was less than 1.45 mm. Therefore, quadrangle thermal shell elements, with a size of 1.45 mm, were applied to mesh the components in this FE simulation, resulting in the element numbers of the blank, punch, die and blankholders being 1320, 8636, 7208 and 5440, respectively, as shown in Figure 5b,c. In addition, the 'hot stamping double-action validation strategy' was employed to simulate the forming, quenching and springback processes. The forming temperature was defined as 450 °C, the stamping speed was 250 mm/s and the blankholding force was 10 kN. Considering that sufficient lubricant was applied, the IHTC and friction coefficient under the lubricated conditions were implemented in the FE model. In addition, the temperature-dependent flow stresses of the martensitic steel determined by Sun et al. [23] were implemented, as shown in Figure 6. In the forming simulation, the hot martensitic steel blank with six degrees of freedom was first positioned on the blankholders, and then compressed by the punch. These three components moved towards the fixed die together, and deformed the martensitic steel blank into the desired shape, followed by quenching. The temperature evolution, thickness distribution and springback of the formed component could thus be subsequently measured.

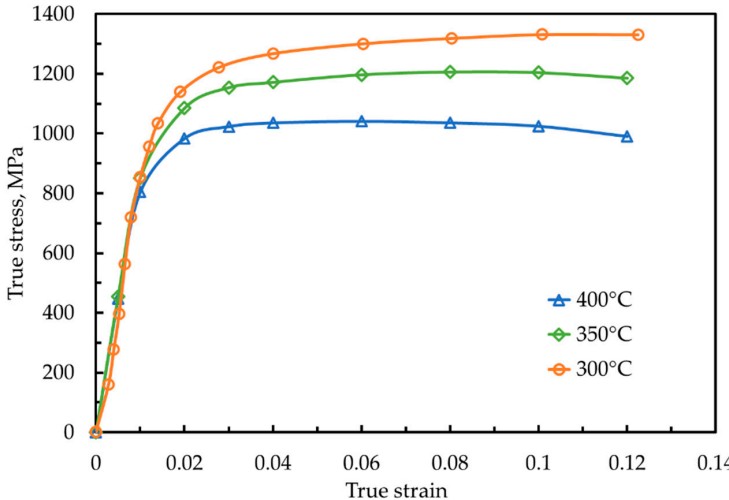

**Figure 6.** The temperature-dependent flow stress of the non-alloy martensitic steel [23].

## 4. Mechanism-Based Models for the Thermomechanical Boundary Conditions

### 4.1. IHTC Model between Martensitic Steel and P20 Steel

An IHTC model was developed to predict the IHTC between the non-alloy martensitic steel and P20 steel under the FAST forming conditions. An overall IHTC was a combined result of three independent heat transfer mechanisms, which were the heat transfers across the air gap, the solid contact and the lubricant layer between the martensitic steel and tools respectively [28], as shown in Equation (1):

$$h = h_a + h_s + h_l, \tag{1}$$

where the air-contact IHTC $h_a$ dominated the heat transfer when a contact pressure and lubricant were not applied. However, $h_a$ was negligible compared to the solid-contact IHTC $h_s$ induced by applying a pressure at the interface and the lubricant-contact IHTC $h_l$ induced by applying a lubricant on the tools. Therefore, $h_a$ was considered as a constant value of 0.15 kW/(m$^2$·K), while $h_s$ was modelled by Equation (2) [28]:

$$h_s = \alpha \frac{K_{st}}{R} N_P, \tag{2}$$

where $\alpha$ was a model constant, $K_{st}$ was the equivalent thermal conductivity of the interface in the unit of W/(m·K), R was the equivalent roughness of the interface in the unit of m and $N_P$ was the pressure-dependent parameter.

The heat transfer across the solid contact interface increased with the increasing thermal conductivity of the blank and tooling. Therefore, the equivalent $K_{st}$ determined by the harmonic mean value of the thermal conductivities of the two contact solids, was used to represent the heat transfer capability of the interface. Meanwhile, the heat transfer across the solid contact interface decreased with increasing roughness of the blank and tooling. Therefore, the equivalent R determined by the root mean squared value of the surface roughness of the two contact solids, was able to describe the surface conditions of the contact interface. The pressure-dependent parameter $N_P$ was the driving mechanism for $h_s$, describing the pressure effect as shown in Equation (3) [28]:

$$N_P = 1 - \exp(-\lambda \frac{P}{\sigma_U}), \tag{3}$$

where $\lambda$ was a model constant, P was the pressure in MPa and $\sigma_U$ was the ultimate tensile stress of the martensitic steel. The term $P/\sigma_U$ was used to describe the pressure mechanism on the real contact area at the interface.

Similarly, $h_l$ was positively affected by the equivalent thermal conductivity of the lubricated contact interface $K_{stl}$ but negatively affected by the equivalent surface roughness R. When a lubricant was applied, the equivalent $K_{stl}$ was the harmonic mean value of the blank, tool and lubricant, while the equivalent R remained. The lubricant-dependent parameter $N_L$ was the driving mechanism for $h_l$, describing the lubricant effect, which was modelled by Equations (4) and (5) [28]:

$$h_l = \beta \frac{K_{stl}}{R} N_L, \tag{4}$$

$$N_L = 1 - \exp(-\gamma \delta_l), \tag{5}$$

where $\beta$ and $\gamma$ were model constants, and $\delta_l$ was the lubricant layer thickness. Therefore, the developed model consisting of Equations (1)–(5), enabled the prediction of the IHTC evolutions for the martensitic steel from 0.5 to 55 MPa for a FAST forming process under both dry and lubricated conditions. The model constants were calibrated using the characterized IHTC results of Section 5.1.1, shown in Table 4, while the modelled IHTC evolutions for the martensitic steel under the FAST forming conditions are shown in Figure 7.

**Table 4.** The constants in the IHTC and friction coefficient models.

| Constant | $h_a$ | $\sigma_U$ | $\gamma$ | $\delta_l$ | $\alpha$ | $\lambda$ | $\beta$ |
|---|---|---|---|---|---|---|---|
| Unit | kW/(m$^2$·K) | MPa | μm$^{-1}$ | μm | - | - | - |
| Value | 0.15 | 90 | 0.15 | 20 | $4.2 \cdot 10^{-4}$ | 6.8 | $9.8 \cdot 10^{-5}$ |
| **Constant** | $\mu_{l0}$ | $\mu_{d0}$ | $Q_l$ | $Q_d$ | $R_0$ | $\varepsilon_1$ | $\varepsilon_2$ |
| Unit | - | - | J/mol | J/mol | J/(mol·K) | - | - |
| Value | 0.0133 | 0.1532 | 14,432 | 6650.7 | 8.31 | 57.25 | 0.61 |
| **Constant** | c | $n_1$ | $n_2$ | $n_3$ | $\eta_0$ | $Q_\eta$ | v |
| Unit | kg/(mm$^2$·s) | - | - | - | mm$^2$/s | J/mol | mm/s |
| Value | 90 | 0.74 | 2.99 | 0.08 | $6.04 \cdot 10^{-4}$ | 3433.6 | 50 |

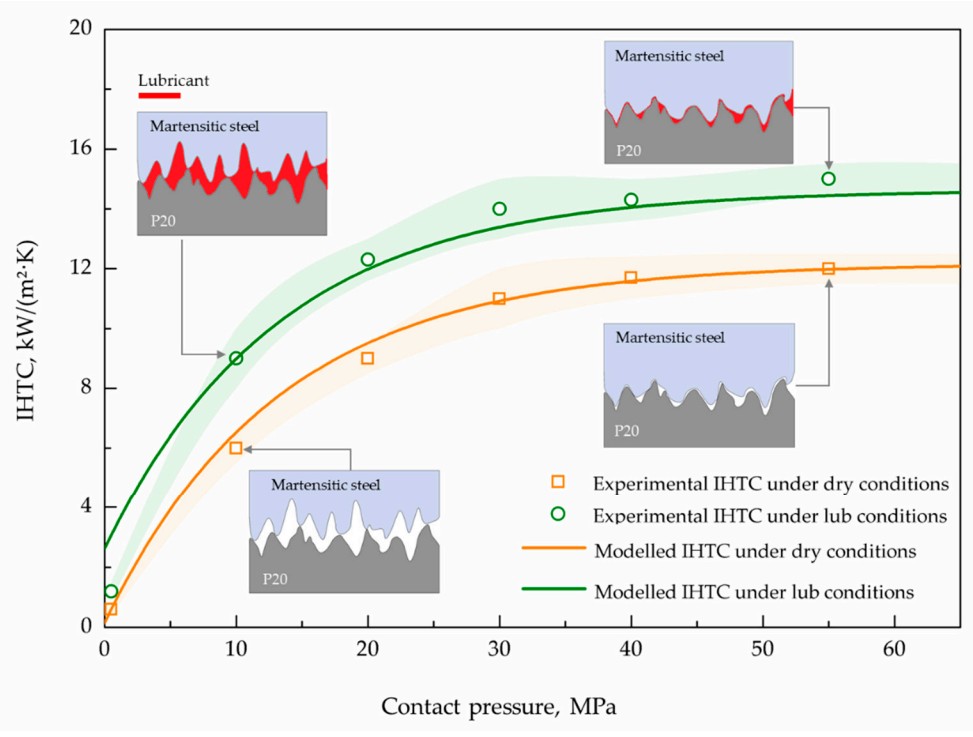

**Figure 7.** The experimental and modelled IHTC evolutions between the martensitic steel and P20 tool steel at a range of pressures, under the dry- and lubricated-contact conditions.

### 4.2. Friction Coefficient Model between Martensitic Steel and P20 Steel

A friction coefficient model was developed to predict the friction coefficient between the non-alloy martensitic steel and P20 steel, considering the effects of sliding distance, temperature and lubricant for a FAST forming process. Similar to the overall IHTC, the overall friction coefficient was a combination of two contributors, stemming from the lubricated-contact friction and dry-contact friction, respectively. Due to the contact condition of the blank–tool interface varying with sliding time/distance in a forming process, the overall friction coefficient was therefore modelled as a time-dependent Equation (6), enabling the prediction of the friction coefficient evolution with sliding distance and time [13].

$$\mu(t) = (1 - \omega)\mu_l + \omega\mu_d, \tag{6}$$

where $\mu_l$ was the friction coefficient under the boundary-lubrication condition, representing a constant value when sufficient lubricant was applied, and $\mu_d$ was the friction coefficient under the dry sliding condition, representing a constant value when the lubricant breakdown occurred. Both $\mu_l$ and $\mu_d$ decreased with increasing temperature, and are thus modelled by Arrhenius equations as temperature-dependent variables, as shown in Equations (7) and (8):

$$\mu_l = \mu_{l0}\exp(\frac{Q_l}{R_0 T}), \tag{7}$$

$$\mu_d = \mu_{d0}\exp(\frac{Q_d}{R_0 T}), \tag{8}$$

where $\mu_{l0}$, $\mu_{d0}$, $Q_l$ and $Q_d$ are model constants, $R_0$ is the universal gas constant and T is the temperature in Kelvin [temperature in degree Celsius (°C) + 237.15]. The contribution of $\mu_l$ and $\mu_d$ to the overall friction coefficient was determined by the lubrication fraction $\omega$, modelled by Equation (9) [13]:

$$\omega = \exp[-\varepsilon_1 \delta_l(t)^{\varepsilon_2}], \tag{9}$$

where $\varepsilon_1$ and $\varepsilon_2$ are model constants, and $\delta_l(t)$ is the instantaneous lubricant layer thickness.

The initial lubricant layer thickness $\delta_l(0)$ resulted in a $\omega$ that approaches 0 and thus $\mu = \mu_l$, representing the boundary lubrication condition at the contact interface. $\omega$ increased with decreasing $\delta_l(t)$ and became 1 when lubricant breakdown occurred, representing the dry condition at the contact interface. Therefore, $\omega$ ranging from 0 to 1 was identified by an exponential function of $\delta_l(t)$, describing the instantaneous lubrication mode evolving from the boundary lubrication condition to the dry condition. The lubricant diminution rate $\dot{\delta}(t)$ was used to calculate $\delta_l(t)$ as Equation (10) [13]:

$$\dot{\delta}(t) = -\delta_l(0)(cP^{n_1}v^{n_2}/\eta^{n_3}), \tag{10}$$

where c, $n_1$, $n_2$ and $n_3$ are model constants, P is the contact pressure, v is the sliding speed and $\eta$ is the lubricant viscosity. $\dot{\delta}(t)$ increased with increasing pressure and sliding speed but decreased with increasing lubricant viscosity, which was dependent on the temperature, as shown in Equation (11):

$$\eta = \eta_0\exp(\frac{Q_\eta}{R_0 T}), \tag{11}$$

where $\eta_0$ and $Q_\eta$ are model constants. As a result, the lubricant diminution and thus the overall friction coefficient, was affected by the temperature. Therefore, the developed friction coefficient model, consisting of Equations (6)–(11), was capable of predicting the evolutions of the friction coefficient with sliding distance, temperature and lubricant for the martensitic steel in a FAST forming process. The model constants were calibrated using the characterized friction coefficient results from Section 5.1.2, shown in Table 4, while the modelled friction coefficients are shown in Figure 8.

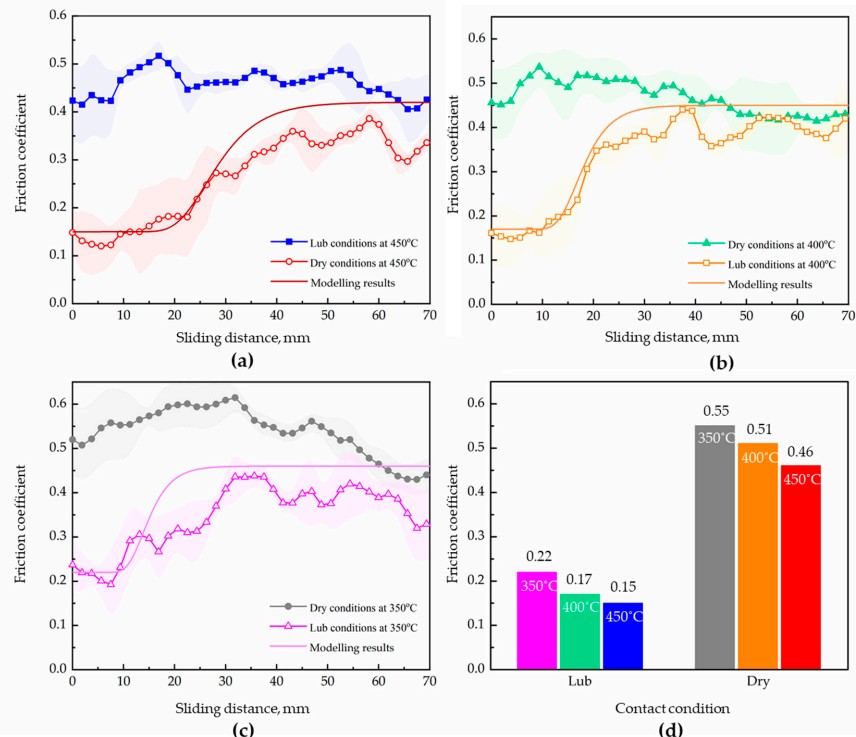

**Figure 8.** The experimental and modelled friction coefficient at a range of sliding distances under the dry- and lubricated-contact conditions, at temperatures of (**a**) 450 °C, (**b**) 400 °C and (**c**) 350 °C; and (**d**) the average friction coefficients at different temperatures under dry- and lubricated-contact conditions.

## 5. Results and Discussion

### 5.1. Thermomechanical Boundary Conditions of Martensitic Steel

#### 5.1.1. IHTC between Martensitic Steel and P20 Steel

Figure 7 shows a close agreement between the experimentally-characterized and numerically-modelled IHTC evolutions for the non-alloy martensitic steel as a function of pressure under the dry- and lubricated-contact conditions. Specifically, the IHTC evolutions under various conditions had an exponentially increasing tendency, that is, a rapid increase in the IHTC followed by a stable value with increasing contact pressure. This was because the real contact area between the hot martensitic steel and cold P20 tool steel was significantly smaller than the apparent area as a result of asperities before compression. However, the contact area was increased significantly by the application of pressure, due to the asperities being deformed, resulting in an increased contacting mesh and thus real contact area. Consequently, the IHTC increased rapidly from 0.6 kW/(m$^2$·K) at 0.5 MPa to 11.7 kW/(m$^2$·K) at 40 MPa. When the asperities on the contact surfaces could not be deformed any more, the real contact area, and thus the IHTC, reached their peak values. The contact pressure of 40 MPa was therefore considered as the convergent pressure.

When the graphite lubricant was applied, the interfacial vacancies were filled, generating a lubricant film. The heat transfer across that lubricant film was enhanced due to the large thermal conductivity, resulting in an average increase of the IHTC by 43%, and the increase of the peak IHTC by 25%, from 12 to 15 kW/(m$^2$·K), as demonstrated by the IHTC results under lubricated-contact (abbreviated as lub) conditions, shown in Figure 7. The effect of lubricant was found to decrease with increasing pressure, due to the generated lubricant film, which was reduced with increasing contact pressure. The characterized IHTC for the martensitic steel at a range of pressures under both dry and lubricated conditions was then input into the FE model of the FAST forming process, in order to precisely simulate the temperature distribution, and thus interact with the mechanical field.

### 5.1.2. Friction Coefficient between Martensitic Steel and P20 Steel

A good agreement between the experimentally-characterized and numerically-modelled friction coefficients, as a function of sliding distance and temperature under the dry- and lubricated-contact conditions, was found, as shown in Figure 8. Specifically, the friction coefficient between the non-alloy martensitic steel at 450 °C and P20 tool steel at room temperature fluctuated, at approximately 0.46, within the sliding distance of 70 mm under the dry sliding conditions, as shown in Figure 8a. The average friction coefficient increased slightly, to 0.51 and 0.55, when the temperature decreased to 400 °C and 350 °C, respectively. The zinc coating adhered to the contact surface of the martensitic steel and acted as a viscous lubricant, transitioning the friction mechanism from interface-sliding to intra-film when the temperature increased [29]. Therefore, the friction force stemmed from the shear strength of the zinc coating, which decreased with increasing temperature. As a result, the friction coefficient between the martensitic steel and P20 tool steel was found to slightly decrease, from 0.55 to 0.46, with increasing temperature, from 350 °C to 450 °C under the dry conditions.

When the graphite lubricant was applied at 450 °C, the friction coefficient fluctuated, at approximately 0.15, during sliding for the first 20 mm, as demonstrated by the friction coefficient evolution under lubricated-contact (abbreviated as lub) conditions in Figure 8a. The contact between the hot martensitic steel and cold P20 tool steel was separated by the applied lubricant film, of which the shear strength contributed to the low friction force and thus friction coefficient [30]. When the sliding distance increased from 20 to 70 mm, the friction coefficient increased significantly, from 0.15 to 0.35. As the lubricant diminished, the solid contact between the two steels significantly increased at the interface, deforming (and/or fracturing) the asperities on the contact surfaces. With the consumption of the lubricant, the friction force consisted of the shear strengths of the remaining lubricant and the deformed (and/or fractured) asperities [31]. The lubricant breakdown would occur when the thickness of the remaining lubricant was lower than the height of the asperities on the tools, leading to a stable friction coefficient close to that under the dry conditions [32]. Meanwhile, it was also found that the evolutions of the friction coefficient followed a similar increasing trend at different temperatures, as shown in Figure 8b,c. However, the lubricant breakdown occurred earlier, at a lower temperature. This is due to the fact that the strength of the martensitic steel and the viscosity of the lubricant decreased with increasing temperature, leading to decreased contact pressure and thus diminution rate of the lubricant thickness. Additionally, the initial values of the friction coefficient still slightly decreased, from 0.22 to 0.15, with increasing temperature from 350 °C to 450 °C, indicating that the effect of temperature on the friction coefficient was consistent for different contact conditions. As mentioned in Section 2.3 sufficient lubricant was used on the tooling to avoid the lubricant breakdown. Hence, the average friction coefficients at different temperatures, under the lubricated conditions shown in Figure 8d, were assigned in the FE model of the FAST forming.

### 5.2. Experimental Validation

#### 5.2.1. Temperature Evolution

In order to validate the determined thermomechanical boundary conditions of the martensitic steel, U-shaped demonstrator components were formed under the FAST forming conditions, and their temperature evolution, thickness distribution and springback were measured and then compared with the FE-simulated results. Figure 9 shows the measured and simulated temperature evolutions of the sidewall (Test point 1) and bottom (Test point 2) of the formed component, respectively. In a FAST forming process conducted at 400 °C, the blankholding area of the martensitic steel was first clamped by the blankholders, and the bottom of the martensitic steel was subsequently deformed by the forming tools, followed by deformation of the sidewalls of the martensitic steel. Therefore, the temperature drop of Test point 1 at the bottom was significantly faster than that of Test point 2 at the sidewall. Compared to the simulated results, the maximum and average errors of the temperature were below 10% and 5%, respectively, indicating the good accuracy of the determined IHTC evolutions. As a result,

the temperature-dependent friction coefficient could be used in the FE simulation to accurately predict the thickness distribution and springback.

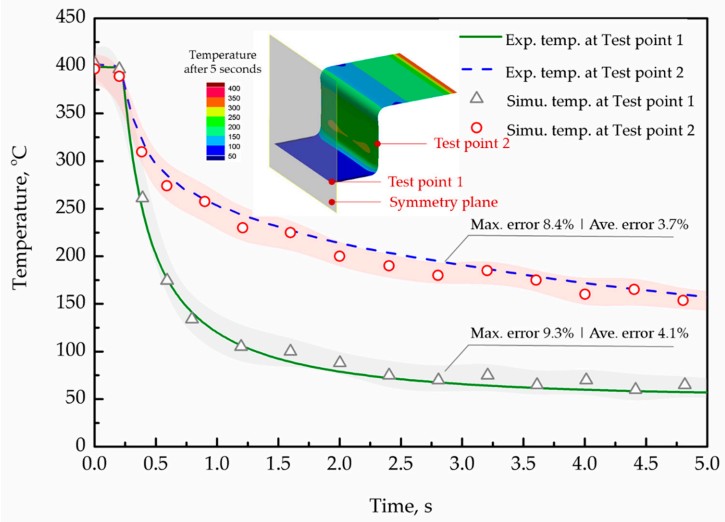

**Figure 9.** The experimentally-measured and FE-simulated temperature histories of the U-shaped component.

5.2.2. Thickness Distribution

Figure 10 shows the experimental and simulated thickness distribution of the central cross-section from the formed component, with the maximum and average errors of 9.5% and 7.7%, respectively. Due to the application of the lubricant, the material was smoothly drawn into the die cavity, resulting in minor plastic deformation and thinning at its blankholding area. After the blankholding process, the bottom of the martensitic steel was compressed between the punch and die, and continuously moved downwards to form the sidewalls. Hence, the plastic deformation of the martensitic steel blank mainly occurred in the sidewalls due to stretching and bending during the draw-in of the material. As a result of the high strength of the martensitic steel, its plastic deformation, and thus thinning, were successfully captured by the simulation. Therefore, the excellent fitting between the experimentally-measured and FE-simulated thickness distribution indicated that the determined IHTC and friction coefficient were accurate in predicting the plastic behavior and drawability of the martensitic steel during the FAST forming process.

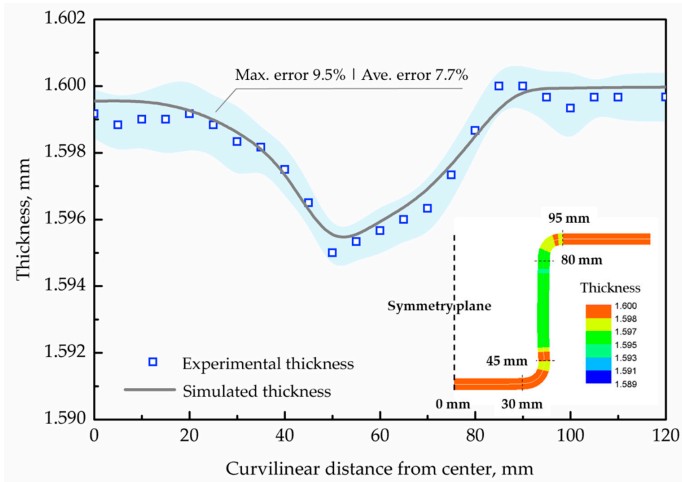

**Figure 10.** The experimentally-measured and FE-simulated thickness distribution of the U-shaped component.

### 5.2.3. Springback

After unloading the press machine, the springback of the U-shaped component formed at 400 °C was characterized by measuring the angles of $\alpha$ and $\beta$, which were approximately 3.28° and 4.61°, as shown in Figure 11. During forming, the blank was drawn into the die cavity after the die (top) corners, and then bent by the die to form the sidewalls of the component, resulting in compressive stress in the upper layer of the blank, and tensile stress in the lower layer. However, the blank at the punch (bottom) corners was subjected to tensile stress in its upper layer, and compressive stress in its lower layer, due to plastic stretching and bending over those corners [33]. The stress gradient between the punch corners and sidewalls of the blank generated a bending moment, leading to the springback angle $\alpha$. The stress distribution at the die corners was similar to that at the punch corners. Consequently, the springback angle $\beta$ was formed due to the stress gradient between the die corners and sidewalls of the component. Furthermore, the residual stress increased with decreasing temperature, leading to the springback angle $\beta$ at the die corners being larger than $\alpha$ at the punch corners. The simulated springback angels $\alpha$ and $\beta$ were 2.96° and 4.87°, respectively, which had errors of 9.6% and 5.4% compared to the experimental results. In addition, the comparisons between the experimental and simulated springback angles, at a range of temperature from 25 °C to 450 °C, are also shown in Figure 11. The close agreements prove the accuracy of the determined friction coefficient.

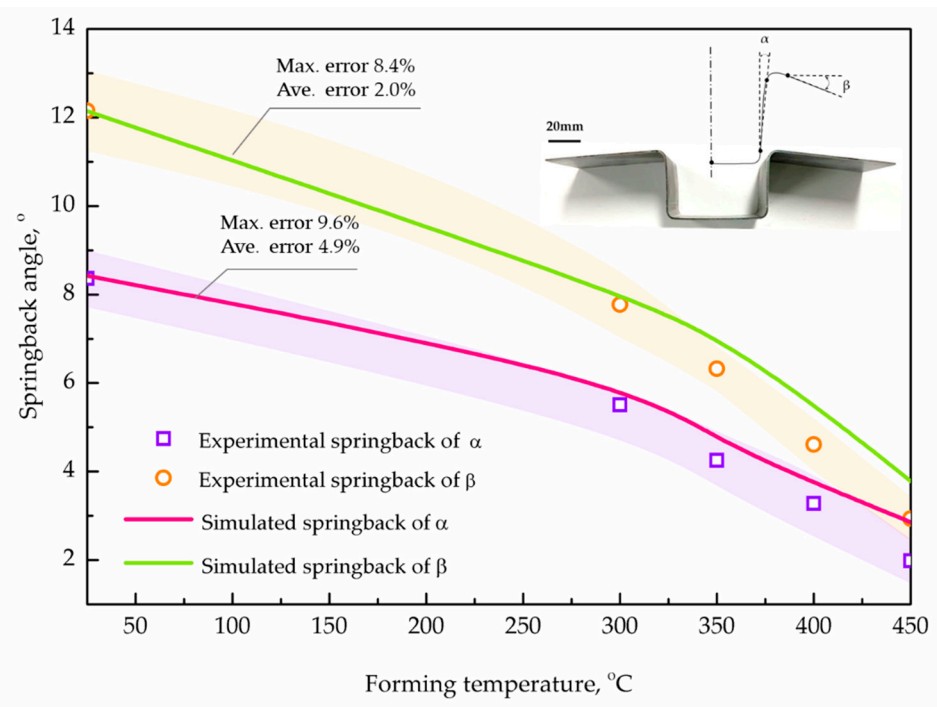

**Figure 11.** The experimentally-measured and FE-simulated springback angles of the U-shaped component.

Therefore, the IHTC and friction coefficient for the martensitic steel characterized in the present research were successfully verified by the comparisons between the experimental and simulated temperature evolution, thickness distribution and springback of the U-shaped component formed under the FAST forming conditions. As a result, the developed FE simulation of the FAST forming process was able to achieve predictive accuracy of the FE models for non-isothermal processes, with an error of less than 10%.

### 6. Conclusions

The thermomechanical boundary conditions of materials are critical to the successful FE simulation of non-isothermal forming processes. The present research characterized and modelled

the thermomechanical boundary conditions of the non-alloy martensitic steel, including the interfacial heat transfer coefficient and friction coefficient under the FAST forming conditions, which were subsequently verified by the experimental results of the FAST forming of U-shaped demonstrator components. The detailed findings of this work are summarized below:

- The IHTC for the martensitic steel rapidly increased at the initial stage and reached a stable value after 40 MPa, for either the dry or lubricated-contact conditions. The application of the graphite lubricant could increase the IHTC by 43% on average.
- The friction coefficient for the martensitic steel was identified at elevated temperatures, from 350 °C to 450 °C, for both dry- and lubricated-contact conditions, showing that the friction coefficient slightly decreased with increasing temperature, and was significantly decreased by the application of the graphite lubricant.
- Mechanism-based IHTC and friction coefficient models for the martensitic steel were developed to predict the IHTC at a range of pressures, as well as the friction coefficient at a range of sliding distances and temperatures for dry- and lubricated-contact conditions.
- The agreements between the experimentally-measured and FE-simulated temperature evolutions, thickness distributions and springback of the U-shaped components achieved close agreement, with an error of less than 10%, verifying the characterized thermomechanical boundary conditions and developed models of the martensitic steel.

**Author Contributions:** Conceptualization, X.L. and L.W.; methodology, X.L., Y.S. and X.Y.; software, X.L.; validation, X.L. and Y.S.; writing—original draft preparation, X.L.; writing—review and editing, X.L., D.J.P., K.-i.M. and L.W.; supervision, L.W. All authors have read and agreed to the published version of the manuscript.

**Funding:** This research received no external funding.

**Acknowledgments:** The authors appreciated the support from ThyssenKrupp Steel for supplying the test materials and China Academy of Launch Vehicle Technology for providing financial support.

**Conflicts of Interest:** The authors declare no conflict of interest.

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
