# Peer review of "Characterization of Thermomechanical Boundary Conditions of a Martensitic Steel for a FAST Forming Process"

_jmmp, doi:10.3390/jmmp4020057_

Round 1
Reviewer 1 Report
The Authors focused on the characterization and modelling the interfacial heat transfer coefficient and friction coefficient of a martensitic steel for a Fast light Alloy Stamping Technology process. Their models have been validated through temperature evolution, thickness distribution and springback measurements on experimentally formed demonstrator components. The article seems to be interesting for a large number of scholars and engineers. It creates a logical scientific research and that is why in my opinion could be published in "Manufacturing and Materials Processing". Some of the comments on the manuscript are listed below.
1) Line 23 and 24; some keywords have been already used in the title of the manuscript. Please change them into different ones (to avoid the keywords repetition with the words used in the title).
2) Table 3 and 4, Figure 5 (the ordinate); some units for example thermal conductivity or specific heat are not uniquely designated i.e. 24.5 kW/mK means (KW/m)·K. Please change it into KW/(m·K) to be unique.
3) Table 4; if it is possible, please change the standard computer notation (e.g. 4.2e-4) into scientific notation (e.g. 4.2·10-4).
4) In the Reviewer opinion the mesh of the blank should be denser. The denser the blank the better the results can be obtained understood as the further error decreases with the increasing number of elements. Did the authors compare the influence of the number of elements on error in the springback analysis?
5) The Reviewer could not find information about size of used elements, kind of elements, number of elements, and some technological dimensions. Such information should be given to the Readers.
6) Line 85 and 111; The dimensions of samples are given in mm2 (line 85) and in mm (line 111). Please unify them because the Readers might be confused.
7) If the Authors experimentally find the coefficient of friction, the schematic drawing should be also shown to the Readers.
8) Please explain, how did you place the thermocouples on the blank (consider showing it on a drawing). How did you avoid the damage of them during the test and did you obtain the correct measurements (thermocouples are very sensitive to mechanical pressing)?
9) Line 218; what is Kelvin temperature?
10) Equations (2 - 11) and Table 4; if they are taken from the literature, then the citations are needed. Some quantities are given without units e.g. the contact pressure (P).
11) Figure 3; why the Authors are using two different models: solid-shell (a) and shell (b)?
12) The mesh of the punch (Figure 3a) and the punch and blank (Figure 3b) is poorly visible. Please, consider such showing of these parts that their mesh should be clearly presented for Readers.
Reviewer 2 Report
Dear Authors,
I agree with you that the thermomechanical boundary conditions of materials are critical to the successful FE simulation of non-isothermal forming processes. The article is characterized by good structure and composition, and its content is assessed as interesting and worth publishing. While reading the article, I had a comments that, hopefully, will help shape the final version of your article.
- Please add in the abstract and Introduction that the term "martensitic steel" refers to non-alloy steel (as opposed to grades included in the stainless steels group)
- I suggest you add a FEM keyword
- Line 60: it is not clear what limitations are referred to here. Please explain this.
- Line 88: P20 tool steel should be described in more detail.
- Line 91: change „seconds” to „s”.
- Line 92: What thermocouples were used? How were they attached? How were the measurements recorded?
- I think it would be beneficial to add one sentence about the PAM-STAMP program.
- Fig. 4. Add the X axis unit.
- You use the abbreviation/symbol "Lub" in the figures. Although one can easily guess its meaning, please explain it in the text.
- Figure 8: The font with dimensions is too small. By the way, please add spaces before the units (mm).
- References: [5] - please complete bibliographic description; [12] please check if the journal allows for the entry: "under review".
- Editorial comments:
Line 53: add space before [17],
Lines 85 and 90: please remove the square from the unit.
Reviewer 3 Report
Review for Journal of Manufacturing and Materials Processing - 833741
Characterization of Thermomechanical Boundary Conditions of a Martensitic Steel for a FAST Forming Process
The authors address an original idea and an interesting research topic for the journal JMMP: characterization and modelling of the interfacial heat transfer coefficient and friction coefficient of a martensitic steel for a FAST forming process. The paper is clearly written and well organized but, in my opinion, several recommendations should be considered:
Comments and Suggestions for Authors
- Figure 4 represents the true stress vs. true strain. These values are obtained from the stress-strain curve applying two equations based on the hypothesis of constant volume during testing. Thus, the stress-strain data are only collected up to UTS and, consequently, it is impossible to obtain a true stress-true strain curve in which the curve descends. I understand that these data were obtained from a previous published paper of the authors, but in my opinion these data are wrong and, hence, the whole simulation is also wrong.
- More data about FE model is mandatory: software, constitutive model of the material, type of hardening used, etc.
Round 2
Reviewer 1 Report
The Authors made a lot of corrections what increases greatly the value of their manuscript. However, I have one remark concerning the Authors' answer to question number 4. Please, consider giving information to the Readers what would happen with the error if the mesh would be denser (see all the details in the remark 4 in the previous review).
Was the distance between the spherical pin and martensitic steel strip so large during Authors' tests (Figure 2 in the current manuscript)?
Reviewer 2 Report
Dear Authors,
Thank you so much for taking my comments and remarks into account. In Table 1 I propose to give the Fe content in AISI P20 tool steel as "Bal."
Author Response
The authors highly appreciate the careful review. The correction has been made in Table 1.
Reviewer 3 Report
Review for Journal of Manufacturing and Materials Processing - 833741
Characterization of Thermomechanical Boundary Conditions of a Martensitic Steel for a FAST Forming Process
The authors address an original idea and an interesting research topic for the journal JMMP: characterization and modelling of the interfacial heat transfer coefficient and friction coefficient of a martensitic steel for a FAST forming process. The paper is clearly written and well organized but, so far as I know, true stress-strain curves are wrong. Furthermore, authors said that “necking was negligible and insignificant”, which is even less consistent with the decrease presented by the true stress-strain curve at 400 °C. How have the authors obtained these true stress-strain curves?
Sorry, but I cannot recommend this paper for publication until this issue is cleared up.
Round 3
Reviewer 3 Report
Many thanks for clarify this point. Your explanation is clear and it convinced me. I accept the paper in the present form.